# Preparation and Characterization of Edible Dialdehyde Carboxymethyl Cellulose Crosslinked Feather Keratin Films for Food Packaging

**DOI:** 10.3390/polym12010158

**Published:** 2020-01-08

**Authors:** Yao Dou, Liguang Zhang, Buning Zhang, Ming He, Weimei Shi, Shiqing Yang, Yingde Cui, Guoqiang Yin

**Affiliations:** 1Innovation and Practice Base for Postdoctors, Chengdu Polytechnic, Chengdu 610041, Sichuan, China; shiwmcdp@foxmail.com (W.S.); sqyang2004@163.com (S.Y.); 2College of Pharmacy, Suzhou Vocational Health College, Suzhou 215009, Jiangsu, China; ivy0714@126.com; 3College of Materials Science and Engineering, Sichuan University, Chengdu 610041, Sichuan, China; 4Green Chemical Engineering Institute, Zhongkai University of Agriculture and Engineering, Guangzhou 510225, Guangdong, China; eins001@163.com (B.Z.); heming1026@163.com (M.H.); 5Guangzhou Vocational and Technical University of Science and Technology, Guangzhou 510550, Guangdong, China; 13602880087@139.com

**Keywords:** feather keratin, dialdehyde carboxymethyl cellulose, crosslinking, edible film

## Abstract

The development of edible films based on the natural biopolymer feather keratin (FK) from poultry feathers is of great interest to food packaging. Edible dialdehyde carboxymethyl cellulose (DCMC) crosslinked FK films plasticized with glycerol were prepared by a casting method. The effect of DCMC crosslinking on the microstructure, light transmission, aggregate structure, tensile properties, water resistance and water vapor barrier were investigated. The results indicated the formation of both covalent and hydrogen bonding between FK and DCMC to form amorphous FK/DCMC films with good UV-barrier properties and transmittance. However, with increasing DCMC content, a decrease in tensile strength of the FK films indicated that plasticization, induced by hydrophilic properties of the DCMC, partly offset the crosslinking effect. Reduction in the moisture content, solubility and water vapor permeability indicated that DCMC crosslinking slightly reduced the moisture sensitivity of the FK films. Thus, DCMC crosslinking increased the potential viability of the FK films for food packaging applications, offering a value-added product.

## 1. Introduction

Microplastics are plastic fragments, synthetic fibers, and plastic particles of less than 5 mm in diameter, that have been detected in seawater, fresh water, food, air, and even human feces [1,2,3,4]. Microplastics with a large specific surface area that can carry other contaminants, that have potential negative effects on the ecological environment and human health [5], and there is therefore an urgent need to curb the rising trend of global microplastic pollution. To achieve this, researchers have increased the development of environmentally-friendly and fully biodegradable bioplastics.

Various studies have been conducted on the use of protein-based plastics in food packaging due to the abundance, biodegradability and low toxicity of proteins, including gelatin [6], soy protein isolate (SPI) [7], whey protein [8], and wheat gluten [9] derived from agricultural sources. One of the best protein sources is feather keratin (FK), which comes from inedible renewable sources and is available in large quantities from the poultry industry. It is estimated that ~5 billion pounds of feathers are generated as a byproduct in the poultry industry annually worldwide [10]. Although FK is environmentally friendly and inexpensive, its application in food packaging is hindered due to its brittleness in relation to other proteins. Therefore, FK must be modified to meet food packaging requirements by means of plasticizing, crosslinking, grafting, or mixing with other polymers. Currently, the addition of plasticizers is the most common method to improve the flexibility and processability of protein-based films, regardless of the processing method used. However, the addition of plasticizers increases the moisture sensitivity of protein-based films, which is not suitable for the packaging of water-rich foods. Small molecular aldehyde crosslinking agents, such as glutaraldehyde, glyoxal and formaldehyde, have received increasing attention due to their efficient crosslinking of protein-based films, which can significantly improve their water resistance, thermal stability, and barrier properties [11,12,13,14,15,16]. However, the largest drawback of small molecule aldehyde is their toxicity, especially when the produced protein-based films which are to be applied in food packing [17]. Therefore, various studies have focused on the use of natural biomacromolecules as crosslinking agents for the modification of edible protein-based films.

Dialdehyde carboxymethyl cellulose (DCMC) and dialdehyde starch (DAS) are two common polysaccharide oxide crosslinking agents used in protein-based film modification, mainly due to their non-toxicity and good film-forming performance. DCMC is a polymer dialdehyde obtained by the controlled periodate oxidative cleavage of the C2–C3 bond of the anhydroglucose units of carboxymethyl cellulose (CMC) [18]. The presence of aldehyde groups and other polar groups in the structure allows DCMC to form covalent bonds and hydrogen bonds with proteins, such as gelatin [19,20], sericin [21], and SPI [22], to improve the properties of protein-based films. However, in the majority of research concerning the use of edible keratin-based films to pack food products, no DCMC-crosslinking during film formation is applied.

The aim of this work was to determine the influence of DCMC concentration in the film-forming solution on the microstructure, aggregate structure, light transmission, tensile properties, water resistance, and water vapor barrier properties of the FK/DCMC films. Moreover, in this work, the extraction of FK, the preparation of DCMC, and the casting process of composite materials are based on our previous reports [23,24,25]. Considering that the produced films are composed of biopolymers, these are expected to be fully biodegradable.

## 2. Materials and Methods

### 2.1. Materials

Carboxymethyl cellulose sudiom (CMC) was purchased from Aladdin Ltd. (Shanghai, China). Urea, sodium sulfide nonahydrate (Na_2_S·9H_2_O), sodium dodecyl sulfate (SDS), sodium periodate and hydrochloric acid were purchased from Guangzhou Chemical Ltd. (Guangzhou, China). The chicken feathers were collected from farmer’s markets.

### 2.2. Preparation of Feather Keratin Solution

After cleaning and disinfection, the chicken feathers were hydrolyzed by reduction method as described in our previous work [23,24,25]. The feathers were steeped with a 7 mol/L urea solution (1:15, *w/v*) at 50 °C for 24 h. Then, Na_2_S·9H_2_O (40 wt % of the feather) and SDS (1 wt % of the feather) were added to the mixture and the extraction was done at 50 °C for 30 min under constant stirring. The extraction solution was then centrifuged at 1000× *g* for 15 min, and the supernatant was adjusted to neutral using a 1 mol/L hydrochloric acid solution. The neutral solution was desalinated by dialysis using a dialysis bag in distilled water at 5 °C for 72 h (Oso-T8280, 8000–14,000, Union Carbide, Houston, TX, USA). Then, the dialyzate was stored at 5 °C before usage. The dialyzate FK concentration was 2 wt %, measured by applying the dry weight method. 

### 2.3. Preparation of Dialdehyde Carboxymethyl Cellulose

DCMC was prepared using a method similar to that by Mu et al. [20]. CMC (4 g) was dispersed and swollen in 80 mL of distilled water in a beaker at 25 ± 1 °C for 12 h. Subsequently, 40 mL periodate solution (0.11 g/mL) were added to the CMC solution under stirring and the pH was adjusted to 3.0 using a hydrochloric acid solution (1 mol/L). The reaction solution was stirred in the dark at 35 ± 1 °C for 4 h followed by the addition of four times the volume of ethanol to the reaction solution. The resulting precipitate was filtered and cross-washed with distilled water and ethanol until all iodate compounds had been removed. The filter cake was vacuum dried at 35 ± 2 °C for 12 h.

### 2.4. Preparation of FK/DCMC Composite Films

The composite films were prepared using similar processes as described in our previous work [24]. The FK solution was adjusted to pH 9.5 with 0.1 mol/L NaOH aqueous solution and then heated at 40 °C for 10 min under continuous stirring. Glycerol was added as the plasticizer at 30% and 40% (based on dry FK weight). DCMC aqueous solution was prepared by adding 2 g DCMC powder to 48 g deionized water. The pH of the solution was adjusted to 9.5 with 0.1 mol/L NaOH solution by continuous stirring at 40 °C for 30 min. Then, a certain volume of FK and DCMC solutions was mixed and adjusted to pH 9.5 with 0.1 mol/L NaOH solution. The mixed film-forming solution was heated at 50 °C for 20 min under continuous stirring. Then, 30 mL the film-forming solution was poured onto a polyethylene petri dish after ultrasonication and dried in an oven at 55 ± 2 °C for 7 h. The obtained film was then stripped carefully and kept for further testing in a desiccator (25 ± 5 °C, 45 ± 10%RH). A series of the FK/DCMC films were coded as GDC m-n, where m and n are the weight percentages of glycerol and DCMC to FK, respectively. For example, GDC30-5 means the glycerol content and the DAS content relative to the weight of FK were 30 wt % and 5 wt %, respectively. The preparation process for FK/DCMC composite films is shown in Figure 1. Variations of the thickness influence the tensile and barrier properties of the packaging film, therefore control of the thickness in producing films by casting method should be taken into account [26,27,28].The thickness of the FK/DCMC films was measured randomly at 10 different positions by a micrometer (accurate to 0.01 mm) and the average value was ranged from 0.09 to 0.15 mm under various conditions studied and applied in the measurements of the tensile properties and water vapor permeability (WVP).

### 2.5. Characterization

Fourier transform infrared (FTIR) spectra of the CMC and DCMC were performed on an IR Spectrum Scanner (Spectrum 100, Perkin-Elmer, Fremont, CA, USA) at a resolution of 4 cm^−1^ in the wave number range from 600 to 4000 cm^−1^. FTIR spectra of FK/DCMC films were measured on the same IR Spectrum Scanner combination with ATR attachment.

Micrographs of the surface and cross-sections of the films were assessed by scanning electron microscopy (SEM) at an acceleration voltage of 20 kV (Quanta 400, Oxford, UK). The cross-sectional samples were obtained by liquid nitrogen to break the films. Prior to the SEM observations, the samples were coated with a fine gold layer.

X-ray diffraction (XRD) patterns of the films were obtained using an X-ray diffractometer (D/max 2200 UPC, RIGAKU, Kyoto, Japan) operated at voltage of 30 kV (30 mA), a scan speed of 6°/min, and a 2θ scan range of 3~40°.

The FK/DCMC films were cut into bars using a sharp knife to a size of 75 mm × 10 mm. The specimens varied in thickness from 0.09 to 0.15 mm. The thickness was measured at ten points and the average was used. The tensile properties of the composite films were measured using a universal testing machine (CMT6503, Shenzhen MTS Test Machine Company Ltd., Shenzhen, China) according to the ASTM standard D882 at a speed of 10 mm/min. The specimens were conditioned at 24 ± 2 °C and 50 ± 5% relative humidity for 48 h before tensile testing. A 500 N load cell was used and a fixture distance was 40 mm. The measurements were carried out in triplicate and average values calculated.

The water vapor permeability (WVP) of the films was measured using a water vapor transmittance tester (Perme W3/030, Labthink Ltd., CHN, Jinan, China) at 38 ± 1 °C with a gradient of 90 ± 1% RH to 0% RH across the film. All the film samples were cut into circles with diameters of 4 cm. The samples varied in thickness from 0.09 to 0.13 mm. The results were averaged from three samples.

The light transmittance of the composite films was measured in the visible range (400–800 nm) using a spectrophotometer (UV-1800, Shimadzu Corporation, Suzhou, China) following the method described by Limpan [29]. The films were cut and placed in a spectrophotometer cell perpendicular to the beam direction. The transparency of the films was calculated using the following Equation:(1)$$\mathrm{Transparency}\;=\;-\;\left(\log\;T_{600}\right)/\chi \;\mathrm{or}\;A_{600}/\chi$$
where *A*_600_ is the absorbance at 600 nm, *T*_600_ is the transmittance at 600 nm, and *χ* is the film thickness (mm). *A* higher transparency value indicates that the film is less transparent.

The moisture content (MC) and solubility of the films were assessed: by weighing three specimens (40 × 10 mm) of each film and then dried in an oven at 70 ± 2 °C for 24 h. The specimens were then cooled in a desiccator for a few minutes and immediately weighed (*W*_1_). With oscillation, the dried specimens were immersed in distilled water at 25 ± 2 °C for 24 h and the filtered debris was dried in an oven at 70 ± 2 °C for 24 h, cooled, and immediately weighed (*W*_2_).

The MC and solubility of the films were calculated by the following Equations:(2)$$\mathrm{MC}=\left[\left(W_{0}-W_{1}\right)/W_{1}\right]\;\times \;100$$
(3)$$\mathrm{Solubility}=\left[\left(W_{1}-W_{2}\right)/W_{1}\right]\;\times \;100$$
where *W*_0_ is the weight of the sample prior to drying, *W*_1_ is the weight of the sample after drying and cooling, and *W*_2_ is the weight of the dried filtered debris. Each film was tested in triplicate.

## 3. Results and Discussion

### 3.1. FTIR Spectra Analysis

According to Mu’s method [20], DCMC was prepared in the solution by controlled oxidation of sodium periodate. The oxidation of CMC by periodate is characterized by the specific cleavage of the C2–C3 bond of glucose residues, which is a typical Malaprade reaction [22]. Such cleavage causes the formation of two aldehyde groups from each glucose unit, forming 2,3-dialdehyde cellulose. The FTIR spectrum of the prepared oxidation products showed a band at 1732 cm^−1^, which is the most characteristic peak for C=O vibrations in the aldehyde groups, as well as a band at 892 cm^−1^, attributed to the formation of hemiacetal between the aldehyde groups and neighbor hydroxyl groups (Figure 2A). The FTIR spectrum is consistent with previous research, which illustrates the generation of aldehyde groups [18,20,21,22].

The FTIR spectra of the DCMC-crosslinked FK composite films were similar and exhibited the typical amide vibrations (Figure 2B). The four bands at ~3280, 1635, 1538 and 1236 cm^−1^ were attributed to the amide A (N–H stretching vibration), amide I (C=O stretching vibration), amide II (N–H bending and C–H stretching vibrations), and amide III (C–N stretching and C=O bending vibrations) bands of FK, respectively [30,31]. The band at 1732 cm^−1^ of the aldehyde group for DCMC in composite films disappeared, indicating the formation of covalent bonds between FK and DCMC. However, the band at ~1660 cm^−1^ for C=N (Schiff’s base) could not be observed, likely due to masking by the strong amide I band of FK [21,32]. Furthermore, the amide A band shifted to a lower frequency with the increase in DCMC content, possibly due to the hydrogen bonding interactions between FK and DCMC [21,32]. Thus, crosslinking and hydrogen bonding occurred between FK and DCMC, which should improve the properties of the composite films.

### 3.2. Film Morphology, Transparency and Aggregate Structure

We previously prepared FK-based films with different compositions [10,23,25,33,34], a slight alkaline environment and polyhydroxy-alcohol as a plasticizer were the key factors to obtain complete, flat, soft, and easily demoulded protein-based films [35]. The slight alkaline conditions were shown to transform the protein peptide chains from winding curls to stretching structures, making peptide chains flexible and exposing more active residues towards interaction with other additives [35,36,37]. Polyhydroxy-alcohol was found to interpose itself between the protein chains and disrupt the interactions between chains, resulting in increased mobility of the protein chains [38]. The combined effect of the above two factors improved the processability of the protein. Herein, both an alkaline environments and glycerol were also shown to have a positive effect on the cross-linking reactions. First, the alkaline environment (pH of 7–10) accelerates the generation of Schiff base between aldehyde and amino groups, and the exposure of more active residues promotes the formation of cross-linking networks [22]. Second, according to the “free volume theory”, plasticizers can improve the flexibility and mobility of proteins, increasing probability of amine group exposure to the far-end aldehyde group of DCMC, thereby facilitating the crosslinking networks formation [20].

As can be observed from our previous work [39], the surface of the FK/CMC composite films appeared grainy and increasingly opaque with increasing CMC content. Herein, the obtained DCMC cross-linked FK films were integrated, smooth, homogeneous, transparent, and flexible. Since the FK solution was light brown, the films also appeared a light brown color. Further, with the increase of DCMC content, these composite films did not show any change in the appearance. However, the surfaces of the films with 40% glycerol had an “oilier” appearance than those with 30% glycerol. This was also observed in the DAS crosslinked FK films [24].

The SEM images of the surface and the cross-section for the unmodified control film (GDC40-0) and DCMC crosslinked film (GDC40-5) exhibited a smooth structure, except for the presence of a few particles arising from the crystallization of salts during drying and impurities from the storage environment (Figure 3A,B). The cross-section of the control film (Figure 3C) and the crosslinked film (Figure 3D) both exhibited a homogeneous, dense microstructure with no phase separation or polymer aggregation, contrary to the phenomenon described in FK/CMC films in our previous report [39]. This phenomenon indicated that the crosslinking reaction induced by aldehyde bonds of DCMC likely improved the compatibility between the polymer components.

From a food safety perspective, a good UV-barrier property is indispensable for food packaging materials since strong UV light can accelerate the oxidation and discoloration of food, which is not conducive to its effective preservation. From previous reports, it is known that protein-based films, such as keratin [24,40], gelatin [19,20] and SPI films [22], have good UV-barrier properties due to their high content of aromatic amino acid residues that absorb UV light. The transmission and transparency of the FK/DCMC films at 600 nm are listed in Table 1. The transmission of FK/DCMC films at the UV light rang (200–280 nm) was almost zero, indicating that these films have good UV-barrier properties. In the visible light range (400–800 nm), all the prepared FK-based films showed a high transmission and a low transparency value (<2) at 600 nm, corresponding to a relatively good transparency. The transparency value of the GDC40 series films was smaller than that of the GDC30 series, indicating greater transparency and a dependency of transparency on glycerol concentration [41]. Additionally, FK/DCMC films were less transparent than those prepared from gelatin/DCMC [20] and SPI/DCMC [22], which is likely due to the presence of pigments in chicken feathers. Nevertheless, the results suggest that DCMC-crosslinked FK films can be used as an anti-UV light translucent material for food packaging.

XRD is widely used in the study of the aggregate structure of polymer materials. FK is a typical semi-crystalline polymer with characteristic diffraction peaks at 2θ values of ~9° and 20°, corresponding to the β-sheet microcrystalline region and the β-sheet diffraction region, respectively [24]. The XRD patterns of the FK films crosslinked with 0%, 2%, and 10% DCMC (Figure 4) showed minimal change in the degree of order in the structure in the film before and after the crosslinking modification. All the DCMC crosslinked FK films were amorphous. Therefore, DCMC crosslinking had a minimal effect on the crystallinity of FK. Thus, changes in the transparency value of FK/DCMC composite films were due to crosslinking between FK and DCMC through the Schiff’s base reaction rather than changes in the crystallinity of FK.

### 3.3. Tensile Properties

Table 2 shows the tensile strength (σ_b_) and elongation at break (ε_b_) of the FK/DCMC films with different DCMC and glycerol content. It was expected that the large molecular DCMC would produce a crosslinking effect similar to that of a small molecule aldehyde crosslinking agent, which has been shown to increase σ_b_ and decrease ε_b_ of protein-based film [15]. However, the composite FK/DCMC films had lower σ_b_ values than the control film both in the GDC30 and GDC40 series films. With the increase in DCMC content, the σ_b_ of the composite film gradually decreased. The ε_b_ values of composite films were higher than that of the control film in the GDC30 series films. Further, the ε_b_ values of composite films were not much different from that of the control film in the GDC40 series films. Similar crosslinking effects were also observed in FK/DAS films [24], gelatin/DAS films [42], zein/DAS films [43,44] and SPI/DAS films [17,45]. This unusual behavior may be due to steric hinerance of large molecules, which prevents DCMC molecules from easily reaching the reactive group in the FK matrix. Moreover, the many hydrophilic groups in DCMC that can adsorb moisture leads to plasticization of water molecules to the film, cancelling part of the crosslinking effect by DCMC. Plasticization by water promotes the mobility of the protein chains, which reduces the tensile strength and improves the ductility of the film. Thus, the indirect plasticization of DCMC is greater than its crosslinking effect on the tensile properties of FK film. In addition, at any DCMC concentration, with the glycerol content increased from 30 to 40 wt %, the σ_b_ of composite film decreased and ε_b_ increased. This results is due to glycerol being a polyhydroxy-alcohol with small molecules, easily entering the protein matrix to interact with polar amino acids and form hydroxyl bonding, which could alter the forces holding the protein chains together and add to the free volume between them. Thus, the mobility of protein chains is improved and the elongation of protein-based films increases with glycerol concentration [6,36,38].

### 3.4. Water Resistance

As a food packaging material, protein-based films must have a certain degree of water stability, especially for the packing of food with high moisture content. To evaluate the effect of DCMC crosslinking on water resistance of composite FK/DCMC films, the MC and solubility of these films were assessed. With the increase in DCMC concentration, the MC of the FK/DCMC films showed a slight decrease followed by a slight increase (Table 2). Overall, the concentration of DCMC had no obvious effect on the MC of the composite films, and its MC value was of ~20%. For the crosslinking effect induced by small molecular aldehydes, the crosslinking network structure formed in the protein-based film could bind the hydrophilic group in the protein, thus considerably reducing the water content of the film. However, the DCMC polymer contains a large amount of hydroxyl groups and is therefore hydrophilic. Thus, DCMC exerts both a crosslinking effect and hydrophilic properties simultaneously, leading to irregularities in the MC of composite films.

As shown in Table 2, with an increasing DCMC concentration, the solubility of the composite films decreased and then increased, reaching ~50%, which was lower than that of the control film in both GDC30 and GDC40 series films. Soluble substances in the composite films may consist of glycerol, small molecular peptides, and DCMC molecules not involved in the crosslinking network. Overall, the GDC40 series films had a higher MC and solubility than the GDC30 series films, likely due to the higher glycerol content in the GDC40 series films [36]. Glycerol is considerably hydrophilic, resulting in the poor water resistance of the film containing more glycerol. Therefore, under the premise of ensuring the processing of protein films, the amount of added glycerol should be kept at minimum.

### 3.5. Water Vapor Permeability (WVP)

To ensure that food is properly hydrated during storage, it is necessary to assess the WVP of packaging films. Table 2 shows the WVP of the composite FK/DCMC films, ranging from 3.3 × 10^−10^ to 5.0 × 10^−10^ g m/m^2^ s Pa, which is higher than that reported for casting gelatin/DCMC film containing 30% glycerol and 10% DCMC (2.1 × 10^−10^ g m/m^2^ s Pa) [20] and lower than that reported for casting wool keratin films plasticized with 30% sorbitol (8.10 × 10^−10^ gm/m^2^ s Pa) [46]. The WVP is mainly affected by the hydrophilic properties of polymer components, flaws in a material, and the internal tortuosity in the microstructure of material [16,47]. With the increase in DCMC concentration, the WVP of the composite films decreased initially and then increased. Furthermore, the WVP of the DCMC crosslinked FK films were all lower than that of the control film both in GDC30 and GDC40 series films. Similar results were also reported in the WVP of SPI/DAS films [17], gelatin/DCMC films [20], and SPI/DCMC films [22]. This change is the result of a combined effect of the following two factors. The crosslinking effect induced by DCMC produces a network structure that increases the density of the film microstructure, thus impeding the diffusion of water molecules within the film. Conversely, the large number of hydrophilic groups in DCMC beneficial to the adsorption of water molecules, increase the diffusion rate of water molecules within the film. In addition, the WVP of GDC40 series films was higher than that of the GDC30 series, indicating that the strong hydrophilicity and free volume effect of glycerol have a great effect on the WVP of protein-based films [44,46,48]. Previous studies have shown that hydrogen bonds can be easily formed between glycerol and proteins due to the small steric hindrance of glycerol, which enables it to enter the protein matrix [35,36].

## 4. Conclusions

The glycerol-plasticized FK films crosslinked by a novel crosslinking reagent DCMC were successfully prepared by the casting process. The FTIR spectra, morphology, light transmission, and crystallization behaviour analyses of the composite films indicated the formation of covalent bonds between FK and DCMC. Covalent bonds and hydrogen bonds occurred between FK and DCMC, which results in the formation of crosslinking networks in the composite films. With the increase in DCMC concentration, the water resistance and water vapor barrier property of the composite films improved slightly, whereas the tensile strength was significantly reduced. These results were not the same as the common crosslinking effect induced by the small molecule aldehydes, most probably because of the polyhydric polymeric nature of DCMC. In common for plasticizer–glycerol in the protein-based films, the increase in the glycerol concentration increases light transmission and water vapor permeability. In general, composite FK/DCMC films were prepared from inexpensive, biodegradable, nontoxic and environmentally friendly materials and have the potential to be used in food packaging.

## Figures and Tables

**Figure 1 polymers-12-00158-f001:**
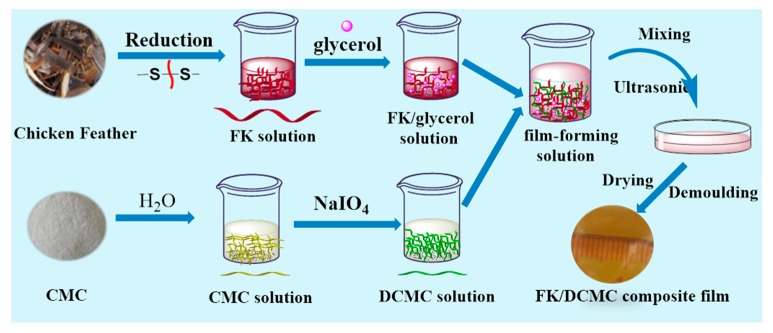
Scheme for the preparation of FK/DCMC composite films.

**Figure 2 polymers-12-00158-f002:**
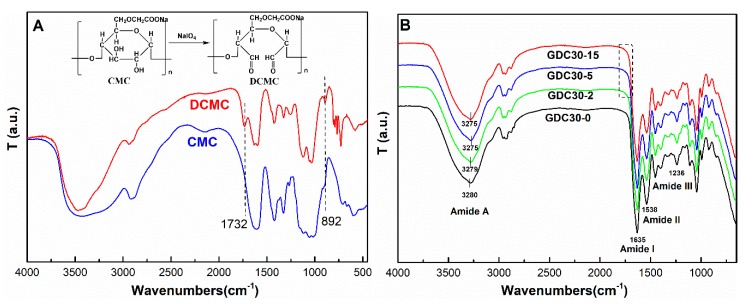
FTIR spectra of (**A**) CMC, DCMC and (**B**) FK/DCMC films.

**Figure 3 polymers-12-00158-f003:**
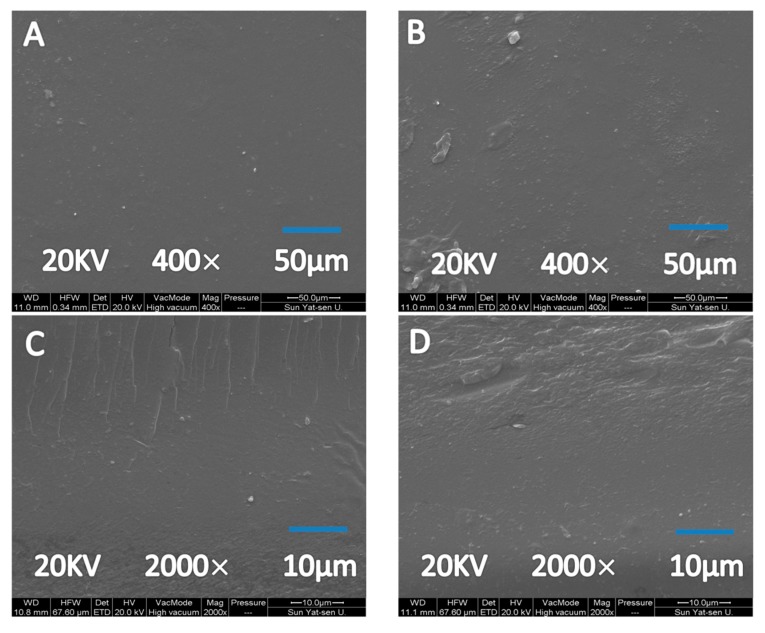
SEM images of surface morphology of GDC40-0 (**A**) and GDC40-5 (**B**), and cross-section morphology of GDC40-0 (**C**) and GDC40-5 (**D**).

**Figure 4 polymers-12-00158-f004:**
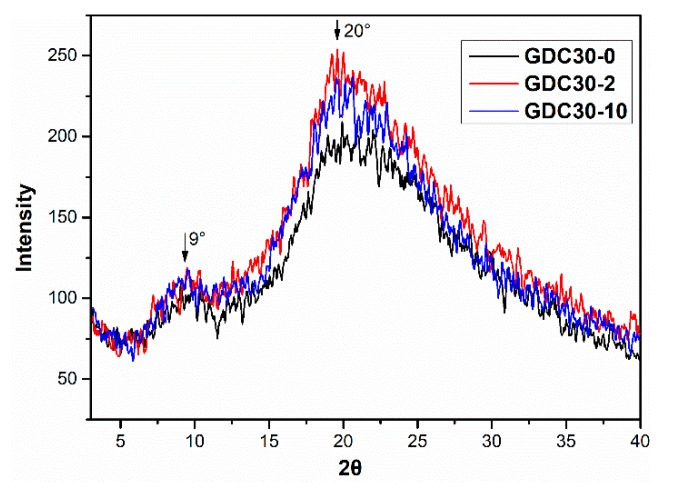
XRD patterns of FK films crosslinked with 0, 2, and 10 (wt %) DCMC.

**Table 1 polymers-12-00158-t001:** Transmission and transparency of the FK films at different glycerol and DCMC contents.

Films			Transmission (%)					Transparency
	200 nm	280 nm	400 nm	500 nm	600 nm	700 nm	800 nm	
**GDC30-0**	0.001 ± 0.001	0.022 ± 0.011	19.14 ± 3.55	38.64 ± 2.37	61.25 ± 2.15	72.21 ± 1.90	77.50 ± 1.54	1.63 ± 0.42
**GDC30-2**	0.001 ± 0.001	0.013 ± 0.009	15.71 ± 2.41	25.51 ± 3.85	53.25 ± 1.76	56.32 ± 2.31	69.46 ± 2.38	1.82 ± 0.22
**GDC30-5**	0.003 ± 0.001	0.041 ± 0.015	13.64 ± 2.73	28.35 ± 5.10	55.86 ± 2.11	69.31 ± 2.42	74.69 ± 1.65	1.81 ± 0.37
**GDC30-10**	0.002 ± 0.002	0.024 ± 0.010	15.36 ± 3.19	17.20 ± 3.14	44.18 ± 3.20	58.78 ± 2.90	65.64 ± 1.74	1.97 ± 0.27
**GDC40-0**	0.001 ± 0.001	0.053 ± 0.021	20.11 ± 3.13	35.37 ± 2.69	60.28 ± 5.45	67.70 ± 1.75	75.38 ± 1.67	1.78 ± 0.59
**GDC40-2**	0.002 ± 0.001	0.031 ± 0.017	19.19 ± 4.27	24.50 ± 2.50	56.35 ± 1.18	72.27 ± 2.12	78.43 ± 2.89	1.23 ± 0.24
**GDC40-5**	0.002 ± 0.002	0.011 ± 0.007	22.56 ± 5.36	39.28 ± 3.37	65.10 ± 2.44	74.59 ± 2.85	77.89 ± 1.30	1.55 ± 0.16
**GDC40-10**	0.001 ± 0.001	0.042 ± 0.023	16.56 ± 4.20	38.74 ± 4.78	62.64 ± 2.59	71.14 ± 2.15	74.06 ± 2.53	1.69 ± 0.35

**Table 2 polymers-12-00158-t002:** Tensile properties, MC, solubility, and WVP of the FK/DCMC films plasticized with 30% and 40% glycerol content, respectively.

Samples	ε_b_ (%)	σ_b_ (MPa)	MC (%)	Solubility (%)	WVP (10^−10^ gm/m^2^ s Pa)
GDC30-0	17.6 ± 3.0	4.0 ± 0.9	20.6 ± 2.1	46.2 ± 4.5	3.7 ± 0.3
GDC30-2	26.8 ± 5.7	2.1 ± 0.8	17.9 ± 0.7	43.8 ± 2.4	3.3 ± 0.2
GDC30-5	28.3 ± 9.1	1.6 ± 0.4	18.7 ± 0.5	42.2 ± 2.1	3.6 ± 0.3
GDC30-10	27.5 ± 3.7	1.2 ± 0.6	19.3 ± 0.8	45.5 ± 3.3	3.8 ± 0.1
GDC40-0	31.0 ± 6.4	3.1 ± 1.1	22.4 ± 1.6	59.2 ± 2.7	5.0 ± 0.6
GDC40-2	29.1 ± 5.1	0.9 ± 0.7	19.5 ± 1.7	48.9 ± 4.3	3.6 ± 0.5
GDC40-5	29.6 ± 2.1	1.0 ± 0.3	18.1 ± 1.2	51.0 ± 3.8	3.8 ± 0.4
GDC40-10	30.8 ± 4.6	0.7 ± 0.4	21.6 ± 0.9	53.3 ± 3.6	4.2 ± 0.2

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
