# Peer review of "Preparation and Characterization of Edible Dialdehyde Carboxymethyl Cellulose Crosslinked Feather Keratin Films for Food Packaging"

_polymers, 2020, doi:10.3390/polym12010158_

Round 1
Reviewer 1 Report
The paper received addresses a topic of interest nowadays. However, I think a number of additions / clarifications are needed. Please, respond to each requirement.
Line 100: the conditions of storage in the desiccator- the temperature and the relative humidity of the air- are missing. Table 1, line 211: the standard deviations related to the results in the table are missing. These should be noted, given that the authors specified triplicate testing of the samples. Also, both in the table and in the rest of the work are not found the notations for films (for example, GDC 30-0 / 2/5/10, GDC 40-0 / 2/5/10). These should be clearly specified. The methods must be fully described so that they can be reproduced. The equations must be completely written, with the description of each element separately. I noticed that the determinations were made according to ASTM D638. Why did you opt for this standard and not the ASTM D882, which is more often used for films' testing? Why didn't you test the thickness of the films? Thickness is closely correlated with solubility, moisture content, optical or mechanical properties and water vapour permeability. An example of this can be observed in the following articlePuscaselu, R.; Gutt, G.; Amariei, S. Biopolymer-Based Films Enriched with Stevia rebaudiana Used for the Development of Edible and Soluble Packaging. Coatings 2019, 9, 360.
Can you determine the applicability of these films? Information about DCMC can be found in the following workJiang, X., Yang, Z., Peng, Y., Han, B., Li, Z., Li, W., Liu, W., Preparation, characterization and feasibility study of dialdehyde carboxymethyl cellulose as a novel crosslinking reagent, Carbohydrate Polymers, 2016, 137, 632-641.
The conclusions should be improved and extended.

Reviewer 2 Report
Thank you for a well written, interesting article.
The experimental part was well designed and the results were appropriately discussed.
Some minor corrections in language:
line 58: replace 'films are' with 'films which are'
line 85: replace 'swellen' with 'swollen'
line 98: change to 'after ultrasonication'
Some comments:
Line 99: how much solution was poured in the dish? this will affect the thickness of the films and might be relevant in a number of experiements (barrier, permability, tensile strength etc)
Tensile strength measurements 3.3:
given the unexpected nature of the results I would expect a much more extensive list of references to be included in attempting to provide an explanation.
Based on line 74-75, some study on biodegradability of the developed films would be appropriate in future work.
Another comment reading the manuscript is that while keratin related references are very recent, the rest of the literature cited in quite old, and therefore not necessarily representative of the latest in knowledge in the field of films. An indicative list of more recent articles that could replace, or be added to, some entries in your reference list are below:
doi:a10.1111/jfpp.12719 pp.1745-1749
International Journal of Biological Macromolecules, 104 (A) 345-359.
doi.org/10.1016/j.foodhyd.2017.05.042
Round 2
Reviewer 1 Report
The paper fulfills the conditions of publication.